# Changes in Mental Health and Preventive Behaviors before and after COVID-19 Vaccination: A Propensity Score Matching (PSM) Study

**DOI:** 10.3390/vaccines9091044

**Published:** 2021-09-19

**Authors:** Yue Yuan, Zhaomin Deng, Musha Chen, Di Yin, Jiazhen Zheng, Yajie Liu, Xinglai Liu, Huachun Zou, Chunhuan Zhang, Caijun Sun

**Affiliations:** 1School of Public Health (Shenzhen), Shenzhen Campus of Sun Yat-sen University, Shenzhen 518107, China; yuany263@mail2.sysu.edu.cn (Y.Y.); dengzhm8@mail2.sysu.edu.cn (Z.D.); chenmsh28@mail2.sysu.edu.cn (M.C.); yind6@mail2.sysu.edu.cn (D.Y.); 429zjz@smu.edu.cn (J.Z.); liuyj237@mail2.sysu.edu.cn (Y.L.); liuxlai@mail2.sysu.edu.cn (X.L.); zouhuachun@mail.sysu.edu.cn (H.Z.); 2Guangzhou Center for Disease Control and Prevention, Guangzhou 510440, China; 3Key Laboratory of Tropical Disease Control (Sun Yat-sen University), Ministry of Education, Guangzhou 510080, China

**Keywords:** COVID-19 vaccine, mass vaccination, health belief, mental health, preventive behavior, propensity score matching (PSM)

## Abstract

Mass vaccination against the COVID-19 pandemic is ongoing worldwide to achieve herd immunity among the general population. However, little is known about how the COVID-19 vaccination would affect mental health and preventive behaviors toward the COVID-19 pandemic. In this study, we conducted a cross-sectional survey to address this issue among 4244 individuals at several COVID-19 vaccination sites in Guangzhou, China. Using univariate analysis and multiple linear regression models, we found that major demographic characteristics, such as biological sex, age, education level, and family per capita income, are the dominant influencing factors associated with health beliefs, mental health, and preventive behaviors. After propensity score matching (PSM) treatment, we further assessed the changes in the scores of health belief, mental health, and preventive behaviors between the pre-vaccination group and the post-vaccination group. When compared to individuals in the pre-vaccination group, a moderate but statistically significant lower score was observed in the post-vaccination group (*p* = 0.010), implying possibly improved psychological conditions after COVID-19 vaccination. In addition, there was also a moderate but statistically higher score of preventive behaviors in the post-vaccination group than in the pre-vaccination group (*p* < 0.001), suggesting a higher probability to take preventive measures after COVID-19 vaccination. These findings have implications for implementing non-pharmaceutical interventions combined with mass vaccination to control the rebound of COVID-19 outbreaks.

## 1. Introduction

Coronavirus disease 2019 (COVID-19) continues to spread worldwide, and the emergence of severe acute respiratory syndrome coronavirus 2 (SARS-CoV-2) variants further brings new challenges in the prevention and control of the global pandemic [1,2]. Although this pandemic has been effectively controlled by non-pharmaceutical interventions (NPIs), including social isolation, mask use, and case isolation in China [3], it remains extremely vulnerable to imported SARS-CoV-2 transmission [4,5]. Mass vaccination against SARS-CoV-2 infection is thought as the most cost-effective strategy to establish a herd immunity barrier and eventually stop this pandemic. So far, more than 17 kinds of COVID-19 vaccines have been approved for clinical use [6].

As of 12 September 2021, more than 2.14 billion doses of COVID-19 vaccine had been administrated in China [6]. Currently, several COVID-19 vaccines have been approved for clinical use [7], but the long-term surveillance of their safety should be further studied, especially among vulnerable populations with medical conditions. In addition, it is known that the effectiveness of current COVID-19 vaccines might not be 100% [8,9], and it will be further compromised as the SARS-CoV-2 variants facilitate the immune escape from the current COVID-19 vaccines [10]. Thus, we should pay attention to the breakthrough infections after vaccination in recent real-world evidence [11,12,13,14]. A noteworthy issue is that the daily preventive behaviors of the general population might change after vaccination [15]. The maintenance of NPIs is necessary to prevent the rebound of the COVID-19 pandemic, until sufficient vaccination coverage is reached for herd immunity [4]. Thus, there is an urgent need to understand the changes in personal preventive behaviors before and after COVID-19 vaccination, which represents a top priority to adjust the prevention and control strategies with the process of mass vaccination.

Another remarkable issue is that the COVID-19 pandemic may affect the mental health status in different countries [16]. For example, clinically generalized anxiety, depressive symptoms, and poor sleep quality among the general population have been proven to be more prevalent during the COVID-19 pandemic in China [17]. However, it is not known whether the mental health status would be affected after COVID-19 vaccination. One study showed that timely HPV vaccination could effectively alleviate anxiety and depression [18], while another study indicated that HPV vaccination is not associated with physical and mental health complaints [19]. In addition, influenza vaccine had a greater negative effect on patients with depression and anxiety than on mentally healthy individuals [20]. Consequently, it is of great significance to investigate how this mass vaccination against COVID-19 would have an impact on mental health among the general population.

In the present study, we conducted a cross-sectional survey to evaluate the effects of COVID-19 vaccination on preventive behaviors and mental health status among the general population in Guangzhou, China. This work will provide insights into adjusting the corresponding strategies for further vaccination promotion and will be helpful to guide appropriate behaviors against COVID-19 during and after mass vaccination.

## 2. Materials and Methods

### 2.1. Study Design and Participants

This investigation was a cross-sectional study and conducted in a population of 18 to 80 year-olds in Guangzhou, the capital of Guangdong Province in China, from 14 April to 18 May 2021. Convenience sampling was used to collect respondents from four different vaccination sites randomly selected in four districts of Guangzhou. Assuming the proportion of the COVID-19 vaccination coverage rate as 50%, 1067 subjects were required based on the formula N = [Z^2^_1−__α_ × (p) × (1 − p)]/d^2^, with a precision level of 0.03. We increased 20% subjects for possible real-world differences, and therefore a minimum of 1280 participants were required. We set up an electronic questionnaire on the website www.wjx.cn, (accessed on 14 April 2021) an online survey platform, and generated a quick response (QR) code for participants to scan and fill in at the vaccination sites. The inclusion criteria were (1) 18 to 80 year-olds (2) willing to participate in this survey. People with comprehension deficits were excluded. 

### 2.2. Survey Tools

The questionnaire used in this survey consists of four sections: (1) demographic characteristics, including biological sex (male or female), age, education level, family monthly per capita income, frequency of domestic and foreign business trips, and influenza vaccination status; (2) health belief model (HBM) scale; (3) questions evaluating the participants’ mental health status; and (4) items regarding preventive behaviors against COVID-19. The questionnaire is provided in Appendix A.

The HBM scale was previously used to evaluate people’s health beliefs and attitudes toward seasonal influenza vaccination [21], as well as to predict the acceptance of COVID-19 vaccination [22,23,24]. In this study, we modified the HBM scale to evaluate the health beliefs of vaccinated individuals. The HBM scale was adapted from above-mentioned literatures [21,22,23] and contained five dimensions: perceived the susceptibility of COVID-19 (four items), perceived the severity of COVID-19 (five items), perceived the benefits of receiving the COVID-19 vaccine (three items), perceived obstacles to access the vaccine (three items), and motivation to get vaccination (three items). The participants were asked how they agreed or disagreed with each statement, and a 5-point Likert scale was used to score each item, from strongly disagree (1 point) to strongly agree (5 points). Except for perceived obstacles to access the vaccine, which was reverse scoring to improve credibility, all other items were forward scoring. That is, higher scores indicated a greater health belief. 

We assessed the mental health status using the adapted PHQ scale [25], which is a widely used assessment tool for self-reported depression. This scale reflects people’s attitudes toward the COVID-19 pandemic and symptoms, as well as their mood, sleep, and attention symptoms in the past month. The daily preventive behaviors against COVID-19 were measured via a modified 9-item scale developed by previous studies [23,26]. The participants were asked about their daily use of masks, hand washing and disinfection, and social distancing. Both the mental health scale and the preventive behavior scale were scored using a 5-point Likert scoring method, from strongly disagree (1 point) to strongly agree (5 points). Higher scores represented worse psychological conditions or a higher probability to take preventive measures against COVID-19. Considering that the above three scales were modified in this study, we used confirmatory factor analysis to evaluate the reliability and validity of these scales (Appendix A).

### 2.3. Definition of Subgroups

The demographic variables in this study were the following factors: biological sex (male or female), education level (junior high school or below, high school, bachelor, master or above), family monthly per capita income (RMB <5000, RMB 5000–10,000, RMB 10,001–15,000, RMB >15,000), health condition (very good, good, general, poor, very poor), influenza vaccination history in the past 3 years (no vaccination, irregular vaccination, regular vaccination), domestic business trip frequency (at least twice a month, once a month, once every 3 months, once every 6 months), and first shot time of COVID-19 vaccine (waiting to receive the first shot or just vaccinated, 2 weeks ago, 1 month ago, 3 months ago, 6 months ago). Previous studies have indicated that changes in psychological and behavioral performance usually occur after a period of vaccination [27,28,29,30]. As a result, the control group (pre-vaccination) was defined as those who were waiting to receive the first shot or were just vaccinated, while those who had been vaccinated more than 2 weeks were defined as the vaccine treatment group (post-vaccination) in our study.

### 2.4. Data Analysis

To ensure that our questionnaires were credible, we cleaned the data using the following procedures: (1) removed those who did not complete the baseline characteristics (such as age and biological sex) in the questionnaire, (2) excluded those who answered the quality control question incorrectly or filled in the scales incompletely, and (3) excluded those who less than 180 s or more than 3600 s to answer. Cronbach’s α coefficient was used to judge the reliability of the questionnaires. The goodness-of-fit indices, including the root mean square error of approximation (RMSEA), comparative fit index (CFI), and Tucker–Lewis index (TLI), were used to judge the suitability of models. Frequency was used as an indicator to describe categorical variables in general demographic characteristics, while the discrete variables were described by median (M) and percentile values (Q1: 25th, Q3: 75th). Differences in individual baseline characteristics were compared using the non-parametric Mann–Whitney U test or the Kruskal–Wallis H test. Multiple linear regression analyses were applied to test the associations of potential explanatory variables with health beliefs, psychological conditions, and preventive behaviors.

Propensity score matching (PSM) is a strategy to reduce the selection bias in observational studies and offers a solution to achieve balanced groups by matching treatment and a series of baseline characteristics as control units [31]. The pre-vaccination group and the post-vaccination group were paired 1:1 based on the propensity scores using the nearest-neighbor matching method. The standardized mean difference (SMD < 0.10 indicated a negligible difference between the groups) and *p*-value were both used as criteria. Then, single-factor analyses were used to compare the differences in scores for the above-mentioned three scales (HBM scale, mental health scale, and preventive behavior scale) between the two groups. Statistical analyses were performed using SPSS 25.0 (IBM Corporation, New York, NY, USA) and Stata version 16.0 (College Station, TX, USA), and the difference was statistically significant at *p* < 0.050.

## 3. Results

### 3.1. Participant Characteristics

A total of 4244 respondents were recruited in this survey. After data cleaning, we obtained 4086 valid questionnaires, and the questionnaire recovery rate was 96.3%. As shown in Table 1, before grouping, 54.0% of the participants were male, more than half of participants had a bachelor’s degree or above, 40.0% had a family monthly per capita income of RMB <5000, and 36.0% had a monthly income of RMB 5000–10,000. In addition, 94.0% of the participants had not received an influenza vaccine in the past 3 years. Only a small proportion of participants had often gone on a business trip in the past year. Among the 4086 valid participants, 2232 were assigned to the pre-vaccination group and 1854 were assigned to the post-vaccination group according to criteria defined in the Materials and Methods section (Figure 1).

### 3.2. Associations between Demographic Characteristics with Health Belief, Mental Health, and Preventive Behavior before PSM Treatment

The associations between demographic characteristics and the scores of the HBM scale, mental health scale, and preventive behavior scale are represented in Table 2. Statistical differences were found in the scores of the HBM scale and preventive behavior scale among biological sex, education level, family income, and domestic business trip frequency (*p* < 0.050). The score of the mental health scale between males and females were not different (*p* = 0.085), but statistical differences were found among people with different education levels and family incomes. Interestingly, the influenza vaccination history had no impact on the above three scales. The scores of the mental health scale had no difference with the frequency of domestic business trips (*p* = 0.132), but significant differences were found with the frequency of overseas business trips (*p* < 0.001). No difference was found in the scores of the HBM and preventive behavior scales between the pre-vaccination group and the post-vaccination group (*p* = 0.643 and *p* = 0.500, respectively), but the score of the mental health scale in the post-vaccination group was higher than that in the pre-vaccination group (*p* = 0.003). 

Multiple linear regression showed that biological sex, education level, and family income are associated with the score of the HBM scale, while the mental health scale score was associated with age, education level, family income, and vaccination time. In addition, the factors influencing behavior change were age, education level, family income, and vaccination time (Appendix A).

### 3.3. PSM Treatment to Balance the Participant Characteristics between Pre-Vaccination and Post-Vaccination Groups

The distribution of demographic characteristic between the pre-vaccination group (n = 2232) and the post-vaccination group (n = 1854) is shown in Table 3. Before PSM treatment, there were significant differences among the following factors: biological sex, age, education level, and family income (*p* < 0.050). After PSM treatment, the pre-vaccination group and the post-vaccination group were paired 1:1, as described in the Materials and Methods section. As a result, there were demographic-characteristic-matched participants in the pre-vaccination group (n = 1680) and the post-vaccination group (n = 1680) after PSM treatment (Figure 1), and the differences in the above-mentioned demographic characteristics disappeared between these two groups (SMD < 0.1, *p* > 0.050; Table 3).

### 3.4. Analysis of Health Belief, Mental Health, and Preventive Behavior between the Pre-Vaccination and Post-Vaccination Groups after PSM Treatment

After PSM treatment, there was slightly difference in the HBM scale score between the pre-vaccination and post-vaccination groups (62 (57, 66) vs. 61 (56, 65), *p* = 0.018), and the scores of perceived susceptibility and perceived barriers in the HBM scale were also statistically different between the two groups (*p* < 0.001). Of note, the mental health score of the pre-vaccination group (27 (23, 30)) was higher than that of the post-vaccination group (26 (22, 29), *p* = 0.010; Table 4), implying possibly improved psychological conditions after COVID-19 vaccination. Moreover, the scores of the preventive behavior scale in the post-vaccination group were slightly higher than those in the pre-vaccination group (36 (35, 41) vs. 36 (34, 40), *p* < 0.001; Table 4), suggesting a higher probability to take preventive measures after COVID-19 vaccination.

## 4. Discussion

Previous studies have generally used the HBM as an independent or intermediate variable to identify the influencing factors and clues of health behavior adoption [32,33,34]. In this study, we used the HBM as an outcome variable to investigate whether health beliefs would change following COVID-19 vaccination. We aimed to explore the potential differences of HBM items between pre-vaccination and post-vaccination populations, including how they understand the severity and susceptibility of SARS-CoV-2 infection, how they understand the efficacy of COVID-19 vaccination, and what obstacles they encounter in getting vaccination. In our study, a moderate but statistically significant difference was found between the post-vaccination group and the pre-vaccination group, even after PSM treatment. These findings should be reasonable, since all participants in this study were either waiting for vaccination or had been vaccinated at COVID-19 vaccination sites. Therefore, they had a willingness to be vaccinated against COVID-19 and remained similar in health beliefs toward COVID-19. Among the items of the HBM, perceived susceptibility of COVID-19 decreased in the post-vaccination group (*p* < 0.001), reflecting that the participants intended to believe that COVID-19 vaccination could reduce the risk of SARS-CoV-2 infection to some extent, which was consistent with previous findings [35]. Another observation was that the scores of perceived susceptibility scores of COVID-19 were low in both pre- and post-vaccination groups, and the reason might be that the Chinese government has controlled the COVID-19 pandemic at a low level, with only sporadic cases via vigorous prevention and control policies [36,37].

Recent studies have indicated that the COVID-19 pandemic has caused a remarkably negative effect on mental health among the general population [16,17]. However, another critical question of great significance is to investigate how COVID-19 vaccination may influence the negative status of mental health toward the COVID-19 pandemic. Therefore, we planned to address this issue by our adaption scale of mental health. Interestingly, the scores of mental health status in our study were all at a low level before and after vaccination, suggesting that there is a relatively healthy mental status among the general population toward the COVID-19 pandemic in China when compared to other countries [38,39]. This might be partly attributed to the impressive controlling performance against the COVID-19 pandemic by the Chinese government, and the participants were less worried about SARS-CoV-2 infection. As a result, a highly trusted government would be a powerful measure to promote mental health among the general population. This observation may give governments and health authorities some important implications to enhance public credibility.

After PSM treatment, there was a moderate but statistically significant lower score of anxiety and depression symptoms in the post-vaccination group than in the pre-vaccination group (*p* = 0.010), demonstrating potentially improved mental health to defeat COVID-19 after vaccination. These findings were consistent with a study, which showed that HPV vaccination might relieve the anxiety and depression of vaccinated individuals [19]. Thus, our data indicated that the COVID-19 vaccine could not only prevent SARS-CoV-2 infection but also reduce the fear of the COVID-19 pandemic and improve the mental health status of vaccinated individuals.

With the process of mass vaccination, one concern was that the daily preventive behaviors against COVID-19 may be reduced among the general population. To our surprise, compared with the pre-vaccination group, participants in the post-vaccination group had a mildly higher frequency to follow preventive behaviors, including wearing a mask and following social distancing. One possible explanation is that the participants received good health education during the COVID-19 vaccination. In addition, the preventive behaviors in the early phase of the pandemic had been transformed into personal habits and awareness, and the continually mandatory mask wearing and social distancing in public places also facilitated strengthening of the daily preventive behaviors among the general population [40]. However, a stochastic dynamic model study previously suggested that a relaxation of NPIs would raise the reproduction number (Rt) value of SARS-CoV-2 back to 1.5, leading to sustained epidemic growth [4]. Thus, to eventually conquer this pandemic, the persistent promotion of preventive measures is still necessary in the future.

Our study had some limitations. First, the effectiveness of cross-sectional studies in examining causality is limited, and therefore further time sequence studies are needed to verify the reliability of these results. Second, the convenient sampling method and the limited number of vaccinations on-site may reduce the sample representativeness in this study. The participants in our investigation were mostly living in Guangzhou, which has a high population density (2059 people per square kilometer [41]) and the vaccination coverage exceeded 70% among the general population [42], and thus the conclusions may not be generalized to other regions.

## 5. Conclusions

In summary, to the best of our knowledge, this is the first study to report how the mental health and preventive behaviors toward COVID-19 would be affected after COVID-19 vaccination. We found a slightly greater awareness of preventive behaviors and a mildly better mental health status among participants in the post-vaccination group than in the pre-vaccination group. Thus, in addition to the direct effect on disease prevention, we suggest that attention should be paid to the benefit of COVID-19 vaccination on mental health improvement for the subsequent promotion of mass vaccination. In addition, given that this pandemic might exist for quite a long time, governments should combine NPIs with mass vaccination together to control the rebound of COVID-19 outbreaks.

## Figures and Tables

**Figure 1 vaccines-09-01044-f001:**
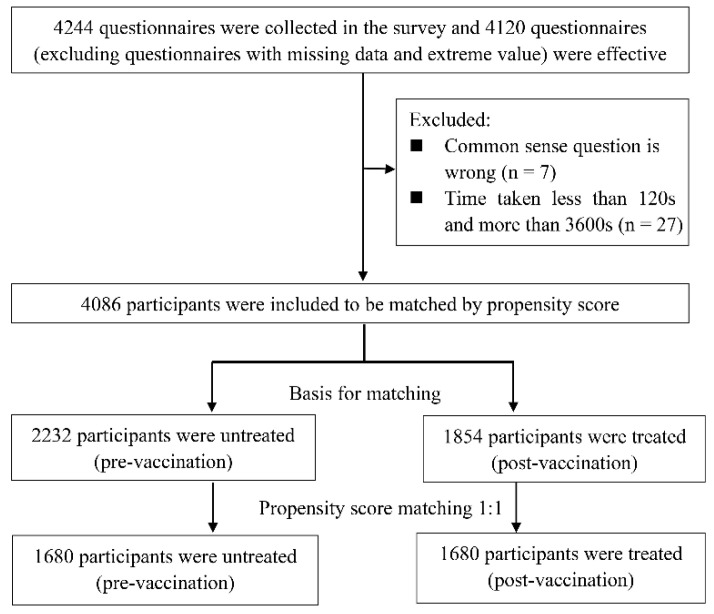
Flowchart of questionnaire collecting and data processing in this study.

**Table 1 vaccines-09-01044-t001:** Demographics characteristics of all samples involved in this survey (n = 4086).

Demographics		N (%) or Mean (SD)
Biological sex		
	Male	2236 (54.7%)
	Female	1850 (45.3%)
Age		34.23 (10.48)
Education		
	Junior high school or below	655 (16.0%)
	High school degree	952 (23.3%)
	Bachelor’s degree	2243 (54.9%)
	Master’s degree or above	236 (5.8%)
Family monthly per capita income		
	RMB <5000	1616 (39.5%)
	RMB 5000–10,000	1491 (36.5%)
	RMB 10,001–15,000	507 (12.4%)
	RMB >15,000	472 (11.6%)
Health condition		
	Very good	2588 (63.3%)
	Good	1295 (31.7%)
	Average	203 (5.0%)
Influenza vaccination status (nearly 3 years)		
	No vaccination	3853 (94.3%)
	Vaccination, discontinuous	194 (4.7%)
	Vaccination, continuous	39 (1.0%)
Domestic business trip frequency		
	At least twice a month	120 (2.9%)
	About once a month	187 (4.6%)
	About once every 3 months	295 (7.2%)
	About once every 6 months	526 (12.9%)
	Barely	2958 (72.4%)
Foreign business trip frequency		
	At least once every 3 months	7 (0.2%)
	About once every 6 months	15 (0.4%)
	About once a year	27 (0.7%)
	Barely	4037 (98.8%)

**Table 2 vaccines-09-01044-t002:** Univariate analysis based on the HBM scale, mental health scale, and preventive behavior scale scores before PSM treatment.

Demographics		HBM			Mental Health			Preventive Behavior		
		M (Q1, Q3)	Z	*p*-Value	M (Q1, Q3)	Z	*p*-Value	M (Q1, Q3)	Z	*p*-Value
Biological sex										
	Male	62 (57, 66)	−2.23	0.026	23 (20, 27)	−1.72 **	0.085	35 (31, 36)	−2.87	0.004
	Female	61 (56, 66)			22 (22, 27)			35 (32, 37)		
Education										
	Junior high school or below	60 (54, 64)	53.39	<0.001	24 (21, 27)	65.20 *	<0.001	36 (33, 36)	69.75	<0.010
	High school degree	61 (56, 66)			23 (20, 27)			36 (33, 37)		
	Bachelor’s degree	62 (57, 66)			22 (19, 26)			34 (30, 36)		
	Master’s degree or above	62 (58, 66)			22 (19, 25)			34 (30, 36)		
Family monthly per capita income										
	RMB <5000	60 (56, 64)	76.62	<0.010	23 (20, 27)	38.35 *	<0.001	35 (32, 36)	17.29	<0.001
	RMB 5000−10,000	62 (57, 66)			22 (20, 26)			35 (31, 37)		
	RMB 10,001–15,000	62 (58, 66)			22 (20, 25)			34 (30, 36)		
	RMB >15,000	63 (58, 67)			22 (19, 25)			34 (30, 37)		
Influenza vaccination status (nearly 3 years)										
	No vaccination	61 (57, 66)	1.46	0.482	23 (20, 27)	4.57 *	0.102	35 (31, 36)	4.91	0.086
	Vaccination, discontinuous	61 (56, 65)			22 (19, 26)			34 (30, 36)		
	Vaccination, continuous	61 (55, 64)			24 (20, 28)			35 (33, 39)		
Domestic business trip frequency (nearly 1 year)										
	At least twice a month	62 (57, 65)	23.79	<0.001	23 (19, 27)	7.08 *	0.132	33 (28, 36)	46.50	<0.001
	At least once a month	62 (57, 66)			23 (20, 26)			34 (30, 36)		
	At least once every 3 months	63 (58, 67)			22 (19, 26)			34 (29, 36)		
	At least once every 6 months	62 (58, 67)			23 (20, 26)			34 (31, 36)		
	Barely	61 (56, 65)			23 (20, 27)			35 (32, 37)		
Foreign business trip frequency (nearly 1 year)										
	At least once every 3 months	58 (54, 69)	0.92	0.821	28 (20, 31)	15.73 *	<0.001	38 (22, 39)	0.67	0.879
	At least once every 6 months	62 (58, 67)			27 (23, 30)			34 (29, 36)		
	At least once a year	59 (55, 66)			24 (21, 27)			34 (31, 36)		
	Barely	61 (57, 66)			23 (20, 27)			35 (31, 36)		
Vaccination										
	Pre-vaccination	61 (57, 65)	−0.46	0.643	23 (20, 26)	−3.01 **	0.003	35 (31, 36)	−0.67	0.500
	Post-vaccination	62 (55, 67)			24 (20, 27)			36 (30, 36)		

Note: * Kruskal–Wallis test; ** Mann–Whitney U test. Vaccination groups were divided into the pre-vaccination group and the post-vaccination group according to vaccination status. Abbreviations: M (Q1, Q3), median (Q1: 25th, Q2: 75th); Z, value of the non-parametric test; HBM, health belief model.

**Table 3 vaccines-09-01044-t003:** PSM treatment to balance the participants’ characteristics between pre-vaccination and post-vaccination groups.

Demographic Characteristics		Before PSM Treatment		SMD	*p*-Value	After PSM Treatment		SMD	*p*-Value
		Pre-Vaccination	Post-Vaccination			Pre-Vaccination	Post-Vaccination		
		n = 2232 (%)	n = 1854 (%)			n = 1680 (%)	n = 1680 (%)		
Biological sex									
	Male	1190 (53.3)	1046 (56.4)	0.06	0.041	912 (54.3)	944 (56.2)	0.02	0.508
	Female	1042 (46.7)	808 (43.6)			768 (45.7)	736 (43.8)		
Age		32.9	35.8	0.28	<0.001	32.50	35.25	0.01	0.822
Education									
	Junior high school or below	286 (12.8)	370 (19.9)	0.28	<0.001	224 (13.3)	330 (19.6)	0.02	0.617
	High school degree	438 (19.6)	514 (27.7)			345 (20.5)	460 (27.4)		
	Bachelor’s degree	1372 (61.4)	872 (47.0)			1056 (62.9)	819 (48.8)		
	Master’s degree or above	137 (6.1)	99 (5.3)			55 (3.3)	71 (4.2)		
Family monthly per capita income									
	RMB <5000	772 (34.6)	845 (45.6)	0.24	<0.001	682 (40.6)	799 (47.6)	0.01	0.715
	RMB 5000–10,000	847 (37.9)	644 (34.7)			651 (38.8)	572 (34.0)		
	RMB 10,001–15,000	308 (13.8)	200 (10.8)			196 (11.7)	160 (9.5)		
	RMB >15,000	306 (13.7)	166 (8.9)			151 (9.0)	149 (8.9)		

Note: Pearson’s chi-square. SMD, standardized mean difference.

**Table 4 vaccines-09-01044-t004:** Analysis of health belief, mental health, and preventive behavior between the pre-vaccination and post-vaccination groups after PSM treatment.

Variable		Vaccination Status		Z	*p*-Value
		Pre-Vaccination (n = 1680)	Post-Vaccination (n = 1680)		
HBM		62 (57, 66)	61 (56, 65)	−2.37	0.018
	Perceived susceptibility	11 (8, 12)	11 (8, 12)	−3.27	<0.001
	Perceived severity	22 (20, 25)	21 (20, 25)	−1.25	0.210
	Perceived benefits	12 (12, 15)	12 (12, 15)	−1.33	0.182
	Perceived barriers	12 (11, 14)	13 (12, 14)	−6.87	<0.001
	Perceived self-efficacy	12 (12, 14)	12 (12, 14)	−3.09	0.002
Mental health		27 (23, 30)	26 (22, 29)	−2.59	0.010
Preventive behavior		36 (34, 40)	36 (35, 41)	−3.67	<0.001

Note: Mann–Whitney U test. HBM, health belief model; Z, value of the test.

## Data Availability

The data that support the findings of this study are not openly available due to being human data and are available from the corresponding author upon reasonable request.

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
