# Peer review of "Changes in Mental Health and Preventive Behaviors before and after COVID-19 Vaccination: A Propensity Score Matching (PSM) Study"

_vaccines, 2021, doi:10.3390/vaccines9091044_

Round 1

Reviewer 1 Report

This study attempts to answer a very important question about the mental health status and behavior of people pre- and post-vaccination. The difference between pre- and post-vaccination for relief maybe significant, but the magnitude is too small. The values are mean +/- SEM or SD? A difference of 0.48 with overlapping error bars is hard to believe will have any meaningful impact. While it is attractive to conclude that vaccines are providing mental relief, given that the data is not robust to support this conclusion, I would be weary of over interpretation of these results.

Specifically, please clearly state in detail the inclusion and exclusion criteria. From the website provided as hyperlink, it is not possible to get a gist of sample questions asked.

Did the mental health questionnaire follow any recommended questionnaires such as SCL-90 or PHQ-9. How was sleep quality measured?

When the authors refer to gender, do they mean biological sex (that is what it appears) or was gender specified by the participants. If specified, please break up the analysis by gender. If only biological sex was assessed (male and female), then please modify the manuscript to clarify that only biological sex was ascertained.

Methods about mental health, sleep quality assessments etc need to be better defined.

Author Response

Point 1: This study attempts to answer a very important question about the mental health status and behavior of people pre- and post-vaccination. The difference between pre- and post-vaccination for relief maybe significant, but the magnitude is too small. The values are mean +/- SEM or SD? A difference of 0.48 with overlapping error bars is hard to believe will have any meaningful impact. While it is attractive to conclude that vaccines are providing mental relief, given that the data is not robust to support this conclusion, I would be weary of over interpretation of these results.

 Response: Thank you for your kind comments. The values we used to compare the scores of scales were mean ± standard deviation (SD). We agree with you that  the difference between pre- and post-vaccination for the changes of mental health is significant but the magnitude is not so robust. As you suggested, we therefore re-described our data with caution to avoid the possible over-interpretation. Such as, “a moderate but statistically significant relief of negative emotions was observed in the post-vaccination group”. In addition, we also re-described the data of preventive behaviors change with caution in the revised manuscript, such as “a higher probability to take preventive measures after COVID-19 vaccination”. We have accordingly modified the related descriptions throughout the revised manuscript as the following.

Line 29-35: When compared to individuals in pre-vaccination group, a moderate but statistically significant lower scores was observed in the post-vaccination group (P=0.010), implying a possibly improved psychological conditions after COVID-19 vaccination. In addition, there was also a moderate but statistically higher scores of preventive behaviors in the post-vaccination group than that of pre-vaccination group (P<0.001), suggesting a higher probability to take preventive measures after COVID-19 vaccination. 

Line 246-247:After PSM treatment, there was slightly difference in the HBM scale score between the pre-vaccination and post-vaccination groups.

Line 254-257: Moreover, the scores of the preventive behaviors scale in the post-vaccination group was slightly higher than that of the pre-vaccination group (P <0.001) (Table 4), suggesting a higher probability to take preventive measures after COVID-19 vaccination. 

Line 267-270:  In our study, a moderate but statistically significant difference was found between the post-vaccination group and pre-vaccination group, even after PSM treatment.

Line 274-278:  Among the items of HBM, perceived susceptibility of COVID-19 was decreased in post-vaccination group (P<0.001), reflecting that the participants were intended to believe that the COVID-19 vaccination could reduce the risk of SARS-CoV-2 infection to some extent, which was consistent with previous findings.

Line 299-302: After PSM treatment, there was a moderate but statistically significant lower scores of anxiety and depressive symptoms in the post-vaccination group than that of pre-vaccination group (P=0.010), demonstrating a potentially improved mental health to defeat COVID-19 after vaccination. 

Point 2: Specifically, please clearly state in detail the inclusion and exclusion criteria. From the website provided as hyperlink, it is not possible to get a gist of sample questions asked.

Response: Thank you for your kind mention and suggestion. We have stated the following inclusion and exclusion criteria in our revised manuscript:

Line98-100: The inclusion criteria were : (1) 18-80 years old; (2) willing to participate in this survey. People with comprehension deficits were excluded.

Line158-163: To ensure that our questionnaires were credible, we cleaned the data using the following procedures: (1) removed those who did not complete the baseline characteristics (such as age, sex, etc.) in the questionnaire; (2) excluded those who answered the quality control question incorrectly or filled in the scales incompletely; (3) excluded those who used the answer time less than 180 seconds or more than 3600 seconds.

The website hyperlink was only a platform to generate an electronic questionnaire, and then we can collect data by scanning Quick Response (QR) code and filling in the questionnaire.

Point 3: Did the mental health questionnaire follow any recommended questionnaires such as SCL-90 or PHQ-9. How was sleep quality measured?

Response: Thank you for your mention. Yes,  we used an adapted PHQ-9 questionnaires. We are sorry for missing the related description of  PHQ-9 in the original version of our manuscript. Now, we have added these information accordingly in the revised Methods section (Line 125-128).

In our investigation, there was a question about sleep quality on the mental health assessment scale:

C3: In the past month, my sleep quality has deteriorated

  • Strongly disagree
  • Tend to disagree
  • Do not know
  • Tend to agree
  • Strongly agree

Point 4: When the authors refer to gender, do they mean biological sex (that is what it appears) or was gender specified by the participants. If specified, please break up the analysis by gender. If only biological sex was assessed (male and female), then please modify the manuscript to clarify that only biological sex was ascertained.

Response: We appreciate your kind reminder, and we apologize for not clearly defining the gender in the original version of manuscript. The gender in this survey is biological sex (male or female). We have mentioned this information in “2.2 Survey tools” of the revised manuscript (Line 103-104).

Point 5: Methods about mental health, sleep quality assessments etc need to be better defined.

 Response: Thank you for your kind suggestion. We have added this information in “2.2 Survey tools” of the revised manuscript (Line 125-139).

At last, all of your comments and suggestions are so helpful to substantially improve our manuscript. We have taken it into account seriously and try our best to address these questions point-by-point. Thank you again for all of your comments and suggestions.

Reviewer 2 Report

I was invited to revise the paper entitled "Changes in mental health and protective behaviors before and after COVID-19 vaccination: a propensity score matching (PSM) study". It aimed to compare patients behaviors against sarscov2 during and post vaccination campaign. After PS matching procedure, Authors compared 3 different scales between non vaccinated and vaccinated patients. Despite the topic is interesting, the paper presented some criticism:

  • Scores are discrete variables so they should be presented as median and interquartile range;
  • Student's T-test cannot be used for discrete variables;
  • linear regression models cannot be used in this case because independent variables are categorical, so Authors should use logistic regression models;
  • Despite Authors used a convenience sample, power analysis should be presented;
  • Table 1 should be included in table 3;
  • Authors cannot assume that after vaccination there is a change in behaviors: these are not paired data. Authors should rephrase explaining that there is a difference in scales before and after vaccination;
  • Results of Cronbach's Alpha should be reported in tables;
  • As footnote, Authors should report in table all statistical tests performed;
  • English language should be revised.
  •  

Author Response

Point 1: Scores are discrete variables so they should be presented as median and interquartile range;

 Response: Thank you for your kind correction. As you said, the scales scores were indeed discrete variables, and we have presented the median and interquartile range to describe the baseline characteristic distribution in the revised manuscript (Line 169-170; Line 374,Table 2).

Point 2: Student's T-test cannot be used for discrete variables;

Response: Thank you for your kind mention. We previously analyzed these data using Student's T-test, because the sample size in our study was large enough and these data could be normal distribution according to the central limit theorem. In addition, some other publications also adopted this statistical method for similar data. Of course, as you suggested, the non-parametric test (Mann Whitney U test and Kruskal Wallis H test) is more often adopted to analyze these discrete variables, and we therefore re-analyzed these data using non-parametric Mann-Whitney U test or Kruskal Wallis H test. We have stated this description in the revised manuscript (Line 170-172; Line 209-225; Line 374, Table 2; Line 381, Table 4).

Point 3: linear regression models cannot be used in this case because independent variables are categorical, so Authors should use logistic regression models;

Response: Thank you for your question. After setting dummy variables, classified variables can be served as independent variables in multiple linear regression analysis, and logistic regression requires that the dependent variable should be classified variable. We use the scale score as the response variable, which is not a classified variable and cannot be used for logistic regression, so we finally choose multiple linear regression. Thank you.

Point 4: Despite Authors used a convenience sample, power analysis should be presented;

Response: Thank you for your valuable suggestion. We have conducted the sample size estimation in the research design stage:  Assuming the proportion of COVID-19 vaccination coverage rate as 50%, 1,067 subjects were required from the formula: N= [Z21-a ´ (p) ´ (1-p)]/d2, with a precision level of 0.03. We increased 20% subjects for possible real-world differences, and therefore a minimum of 1,280 participants were required.  A total of 4,244 respondents were recruited in this survey. After data cleaning, we finally obtained 4,086 valid questionnaires. We have added this information in the revised manuscript (Line 89-93). According to your suggestion, we did a post hoc analysis to get the power of this study. Post-hoc power analysis, using G*Power version 3.1.9.7, showed that a sample size of 4,086 respondents yields a power of 99% at a significance level of 0.05 (two-sided).

Point 5: Table 1 should be included in table 3;

Response:  Thank you for your mention. In our manuscript, Table 1 showed the demographics characteristics of all samples involved in this survey, while Table 3 presented how to screen and balance the participants’ characteristics between pre-vaccination and post-vaccination groups by propensity score matching. Therefore, these two Tables represent different information in different samples, and we did not think they could be merged. Thank you.

Point 6: Authors cannot assume that after vaccination there is a change in behaviors: these are not paired data. Authors should rephrase explaining that there is a difference in scales before and after vaccination;

Response: Thank you for your valuable mention. We agree with you that  the original data obtained from convenience sampling were not paired data, however, we subsequently used the propensity score matching (PSM) method to match and balance the participants’ characteristics between pre-vaccination and post-vaccination groups. As you said, before PSM treatment, there were significant differences between pre-vaccination and post-vaccination groups among the following factors: gender, age, education level, and family income (P<0.050). However, after PSM treatment, the pre-vaccination group and post-vaccination group  were paired as 1:1, and there were demographic characteristic-matched participants in pre-vaccination group (n=1,680) and post-vaccination group (n=1,680) after PSM treatment (Figure 1), and the differences of above-mentioned demographic characteristics were disappeared between these two groups (SMD<0.1, P>0.050). (Table 3).

In addition, as both you and another reviewer suggested, we re-described our data with caution to avoid the possible over-interpretation. Such as, “a moderate but statistically significant relief of negative emotions was observed in the post-vaccination group”, “the scores of the preventive behaviors scale in the post-vaccination group was slightly higher than that of the pre-vaccination group ( P <0.001) (Table 4), suggesting a higher probability to take preventive measures after COVID-19 vaccination”.

We have accordingly modified the related descriptions throughout the revised manuscript as the following (Line 29-35; Line 254-257; Line 309-323).

Point 7: Results of Cronbach's Alpha should be reported in tables;

Response: We appreciate this kind remind. We have attached the results of Cronbach's Alpha in the supplement file. The reliability of each scale construct was evaluated according to Cronbach’s coefficient, which was 0.794, 0.813, 0.835 for HBM, mental health and protective behaviors, respectively (Supplementary Table 1).

Point 8: As footnote, Authors should report in table all statistical tests performed;

Response: Thank you for your reminder. We have added these information as footnotes in the revised Tables.

“Table 2”

Footnote: *, Kruskal Wallis test; **, Mann Whitney U test; Vaccination groups were divided into pre-vaccination group and post-vaccination group according to vaccination status.

Abbreviations: M (Q1, Q3), median (Q1: 25th, Q2: 75th); Z, the value of the non-parametric test; HBM, health belief model.

Table 3”

Footnote: Person’s Chi square; SMD, standardized mean difference.

Table 4”

Footnote: Mann Whitney U test; HBM, health belief model; Z, the value of the test.

Point 9: English language should be revised.

Response: The English language writing has been carefully edited. Thank you.

At last, all of your comments and suggestions are so helpful to substantially improve our manuscript. We have taken it into account seriously and try our best to address these questions point-by-point. Thank you again for all of your comments and suggestions.

Round 2

Reviewer 1 Report

The manuscript is much improved and inclusion of suggestions by the authors are appreciated.

There are three issues that still remain:

  1. Page 2, Lines 53-54: vaccine safety. As written, it appears that vaccine safety is a foregone conclusion. It is not; COVID-19 vaccine safety is being debated and still evaluated. In fact, the numbers of adverse events for some vaccines are not fully understood. The FDA asked Pfizer and Moderna to add warning labels about heart inflammation. This should be clarified in the Introduction.

  1. Efficacy of vaccines- given the large numbers of breakthrough cases as demonstrated by the recent Israeli study, caution should be used in stating that the vaccines prevent infection, as they don’t and their efficacy is falling by the day. The introduction should be revised accordingly.
  1. It would be scientifically appropriate to use the term Biological sex and not gender throughout the manuscript.

Author Response

Point 1: Page 2, Lines 53-54: vaccine safety. As written, it appears that vaccine safety is a foregone conclusion. It is not; COVID-19 vaccine safety is being debated and still evaluated. In fact, the numbers of adverse events for some vaccines are not fully understood. The FDA asked Pfizer and Moderna to add warning labels about heart inflammation. This should be clarified in the Introduction.

 Response: Thank you for your kind suggestion. We have modified the description of the COVID-19 vaccine safety accordingly in the revised manuscript.

Page 2, Line 55-58: The safety of several COVID-19 vaccines had been approved for clinical use [7], but the long-term surveillance of their safety should be further studied, especially among vulnerable populations with medical conditions.

Point 2: Efficacy of vaccines- given the large numbers of breakthrough cases as demonstrated by the recent Israeli study, caution should be used in stating that the vaccines prevent infection, as they don’t and their efficacy is falling by the day. The introduction should be revised accordingly.

 Response: Thank you for your valuable suggestion. We have added the description of the COVID-19 vaccine efficacy accordingly in the revised manuscript.

Page 2, Line 58-62: In addition, it is known that the effectiveness of current COVID-19 vaccines might not be 100% [8,9], and it will be further compromised as the SARS-CoV-2 variants facilitate the immune escape from current COVID-19 vaccines. Thus, we should pay attention to the breakthrough infections after vaccination in recent real-world evidences [11-14].

Point 3: It would be scientifically appropriate to use the term Biological sex and not gender throughout the manuscript.

 Response: Thank you for your kind mention. We have modified the term accordingly in the revised manuscript. (Page 1, Line 24; Page 3, Line 104, 140; Page 4, Line 157; Page 5, Line 200; Page 6, Line 210, 222, 233; Page 10, Line 369, 373).

Reviewer 2 Report

Authors addressed all points raised. The paper is now acceptable for publication.

Author Response

Thank you again for all of your comments and suggestions.